# Environmental Stimuli: A Major Challenge during Grain Filling in Cereals

**DOI:** 10.3390/ijms24032255

**Published:** 2023-01-23

**Authors:** Zhenning Teng, Yinke Chen, Shuan Meng, Meijuan Duan, Jianhua Zhang, Nenghui Ye

**Affiliations:** 1College of Agriculture, Hunan Agricultural University, Changsha 410128, China; 2Shenzhen Research Institute, The Chinese University of Hong Kong, Shenzhen 518057, China; 3Hunan Provincial Key Laboratory of Rice Stress Biology, Hunan Agricultural University, Changsha 410128, China; 4Department of Biology, Hong Kong Baptist University, Hong Kong 999077, China

**Keywords:** light, temperature, water, fertilizer, saline–alkali stress, cereal crops, grain filling

## Abstract

Light, temperature, water, and fertilizer are arguably the most important environmental factors regulating crop growth and productivity. Environmental stimuli, including low light, extreme temperatures, and water stresses caused by climate change, affect crop growth and production and pose a growing threat to sustainable agriculture. Furthermore, soil salinity is another major environmental constraint affecting crop growth and threatening global food security. The grain filling stage is the final stage of growth and is also the most important stage in cereals, directly determining the grain weight and final yield. However, the grain filling process is extremely vulnerable to different environmental stimuli, especially for inferior spikelets. Given the importance of grain filling in cereals and the deterioration of environmental problems, understanding environmental stimuli and their effects on grain filling constitutes a major focus of crop research. In recent years, significant advances made in this field have led to a good description of the intricate mechanisms by which different environmental stimuli regulate grain filling, as well as approaches to adapt cereals to changing climate conditions and to give them better grain filling. In this review, the current environmental stimuli, their dose–response effect on grain filling, and the physiological and molecular mechanisms involved are discussed. Furthermore, what we can do to help cereal crops adapt to environmental stimuli is elaborated. Overall, we call for future research to delve deeper into the gene function-related research and the commercialization of gene-edited crops. Meanwhile, smart agriculture is the development trend of the future agriculture under environmental stimuli.

## 1. Introduction

Rice (*Oryza sativa* L.), maize (*Zea mays* L.), wheat (*Triticum aestivum* L.), barley (*Hordeum vulgare* L.), and sorghum (*Sorghum bicolor* L.) are important cereal crops in most countries, and they contribute over 50% of the world’s total production of cereals in millions of tonnes [1]. As the number of people and livestock continues to increase, the demand for food production has already outweighed cereal yield [2]. Grain filling refers to the processes of cell proliferation and cell expansion, as well as the processes of photoassimilate interconversion and starch accumulation. Grain filling, as the final stage of growth and also the most important stage in cereals’ life cycles, directly determines the grain weight, grain composition, and therefore quality and final yield [3,4,5].

Agriculture is one of the most vulnerable sectors to changes in the climate and is inherently sensitive to climate conditions; therefore, agricultural production is easily affected by climate variability [6]. For the growth process of cereal crops, grain filling is the most sensitive stage to climate variability, which influences the qualitative and quantitative traits of the final yield. The grain filling rate, grain filling percentage, grain weight, and grain quality are significantly limited by deterioration of the agricultural environment. The factors affecting grain filling are low light, extreme temperatures, drought, flooding, and soil salinity [7,8,9,10,11,12]. Therefore, minimizing the agricultural losses caused by the changing climate is urgent and has become a major global concern to ensure food security.

There are two main sources of carbon during grain filling in cereals. One is current carbohydrate production driven by photosynthesis and transported directly to the grain; the other is assimilates redistributed from reserve pools in vegetative tissues [13,14]. Therefore, both carbon sources are affected and hindered by environmental stimuli. Drought and heat stress can reduce the accumulation of certain seed components, mainly starch and proteins, by suppression of enzymatic reactions (starch and protein synthesis); in this way, these factors have a strong influence on grain weight and quality [5]. Like drought and heat stress, soil saline–alkali stress causes grain yield reduction mainly due to the limitation of carbohydrate production and inhibition of activities of starch-synthesis-related enzymes in grains [15,16]. Low light stress significantly reduces the photosynthetic rate, leading to the inhibition of carbon metabolism in crops [17]. In addition, recent studies showed that light is implicated in the regulation of starch synthesis in grains [18]. In the case of nitrogen fertilizers, too much or too little nitrogen can have negative effects on grain filling. Too much nitrogen results in unfavorably delayed senescence and poor grain filling, resulting in a low grain weight and yield constraints. Conversely, a moderate nitrogen application will balance grain filling between superior and inferior spikelets, thereby improving the grain weight of both superior and inferior spikelets [19]. Mild soil drying can also enhance plant senescence and faster and better remobilization of carbon from reserve pools to grains, leading to faster grain filling of both superior and inferior spikelets [3]. It can be concluded that these factors are multi-faceted and complicated in regulating grain filling.

The physiological and molecular mechanisms of grain filling have been studied intensively in recent years, especially under different environmental stimuli. In this work, we describe the current status of the agricultural environment and stress dose–response effect on grain filling in cereals. We also illustrate how environmental stimuli, including low light, extreme temperatures, drought, fertilizers, and saline–alkali conditions, impact grain filling. At the end of this review, we give some advice in response to these environmental stimuli. We hope that these efforts will provide useful information for researchers and breeders to develop stress-tolerant crops and high-efficiency cultivation in the future.

## 2. Environmental Stimuli Influence Grain Filling in Cereals

### 2.1. Agricultural Production Is Taking on Challenges Imposed by Global Environmental Change

An adequate environment is required for normal plant growth. However, the agricultural environment and ecosystems are changing, creating challenges in the development of agricultural production. Environmental stimuli and their results are even more severe during grain filling (Figure 1).

#### 2.1.1. Light

Light is an essential environmental factor that impacts plant morphogenesis and photosynthetic rate and regulates carbon metabolism in plants [9,18,20]. However, low light caused by global dimming and environmental pollution has become a growing worldwide phenomenon over the last few decades [21,22]. According to the United Nations Intergovernmental Panel on Climate Change (IPCC), the global radiation reaching the Earth’s surface increased by 1.3% on average per decade from 1960 to 2000 [23]. Worse still, there has been a decline of more than 6% per decade in most parts of China [24]. Moreover, a decreasing trend has also been observed in India and Southeast Asian countries, where a decrease in irradiation of up to 40% to 60% is experienced during the wet season. Low light intensity usually causes about 50% yield loss [25,26].

#### 2.1.2. Temperature

Extreme temperatures (heatwaves and low temperature) are one of the major challenges of climate change to agriculture and food security, increasing the risk of large-scale crop failure [10,27,28]. Globally, hot temperature extremes have rapidly increased in distribution, number, and size over the past several decades [29,30]. Further, hot temperature extremes have been increasing continuously, and this upward trend shows a larger tendency for the highest extreme events during the so-called hiatus of global warming [31]. This causes yield and quality declines, such as in the case of a wide-spread severe heat event during the grain filling stage in many counties of South China in 2003 and 2013 [32,33]. In recent years, during the flowering period of single-season rice, temperatures have been reported to higher than 35 °C and last for over 20 days [34]. Similarly, low temperature (freezing or chilling temperatures) is another significant environmental factor that can severely impact crop yields and limit crop production in temperate and high-altitude areas [35]; such temperatures are frequently expected in high-altitude areas and during early spring [36,37].

#### 2.1.3. Water

Water is critical to all stages of crop development that limit crop growth and reduce their yield [38]. Nevertheless, flooding and extreme drought have occurred frequently in recent years [12,39,40]. Drought and excessive rainfall were the first and second largest causes of maize production loss in the United States from 1989 to 2016 [41]. Low temperature and overcast weather generally appear together, while drought often occurs together with high temperature. Such types of compound extreme events induce aggravated impacts on agricultural production.

#### 2.1.4. Soil

Soil is an important medium in which crops survive and grow. However, the quality of cultivated soil is particularly concerning. Soil problems, such as soil nutrient loss [42], soil acidity [43], soil salinization [8], and heavy metal pollution [44] threaten national food security; they are also worsening with climate change and unsustainable cultivation practices.

### 2.2. Stress Dose–Response Relationships in Grain Filling

It is generally believed that the process of grain filling is synergically determined by genetic and environmental factors, in which the latter are the key factors [3,47]. Fully understanding the responses of grain filling under environmental stimuli might provide key solutions to problems in agricultural production under unfavorable environments (Table 1).

Light, temperature, water, and fertilizer are the main environmental factors and have adverse effects on grain filling, especially on inferior grains (Figure 1; Table 1). Superior grains located on apical primary branches usually flower earlier and produce larger and heavier grains, while inferior grains located on the proximal secondary branches are pollinated later and generate smaller grains [48]. Compared with superior grains, inferior grains need more time and energy for grain filling. It is for these reasons that inferior grains are more sensitive to environmental stimuli (Figure 1) [49]. For example, low light and shading inhibit starch synthesis in grains and thereby lead to poor grain filling, particularly for the bottom part of the panicle [18,50]. Overuse of chemical fertilizer (especially nitrogen fertilizer) highly influences the grain filling rate and quality of rice grains [19,51]. Heat stress and water deficit stress reduce the grain weight through a reduction in grain growth duration and grain growth rate, which is more serious in inferior grains [27,52,53,54,55]. However, mild soil drying can enhance whole-plant senescence, lead to the remobilization and transfer of assimilates pre-stored in vegetative tissues to grains, increasing the grain filling rate and grain weight of inferior spikelets [3,56,57]. Poor grain filling caused by saline–alkali stress is another key reason for yield loss in many rice cultivars [15,16]. The inhibitory effect by NaCl is also greater in basal spikelets than in apical spikelets [15]. These effects on inferior grain caused by environmental stimuli notably vary from year to year and lead to variations in the grain weight of cereals. It is therefore important to enhance inferior grain yield in adverse environments to improve and stabilize yield.

**Table 1 ijms-24-02255-t001:** Cereal responses to environmental stimuli during grain filling.

Stress Type	Species Name	Dose/Concentration	Results/Impacts	References
Low light	Wheat	100%, 40%, and 10% natural light were applied for 1, 3, 5, and 7 days during the young microspore stage of two wheat cultivars (Henong825, Kenong9204).	Low light intensity, low Gs (stomatal conductance) and chlorophyll content, and damage to the ultrastructure of the chloroplast and photosynthetic system. The grain number in Henong825 and Kenong9204 was reduced by 3.6–33.3% and by 14.2–72.7%, respectively. The leaf photosynthetic rate (Pn) in Henong825 and Kenong9204 was reduced by 4.5–93.9% and 26.4–99.0%.	[17]
Low light	Rice	Control and 53% low light stress (covered by a layer of white cotton yarn screens) from flowering to maturity.	Low light resulted in decreased activity of ADP-glucose pyrophosphorylase (AGPase), sucrose synthase (SuS), soluble starch synthase (SS), granule-bound starch synthase (GBSS), and starch branching enzyme (SBE), eventually leading to reduced maximum and mean grain filling rates. Inferior grains were more severely affected than superior grains.	[18]
Heat stress	Wheat	Heat stress caused by different sowing periods. The late-sown crop experienced 6 ± 8 °C warmer temperatures during grain development than the crop sown at the normal time.	Heat stress reduced both the grain growth duration and the grain growth rate. Starch and protein synthesis were more thermotolerant in developing grains of the heat-tolerant variety (Hindi62) than in those of the heat-sensitive variety (PBW154).	[52]
High temperature stress	Wheat	Moved pots to an artificial intelligence greenhouse with daily average temperatures of 25, 30, 35, and 40 °C during different grain filling periods.	High temperature stress during grain filling decreased the duration of grain filling and grain weight, which increased the grain protein content relatively. Heat damage led to a lower quality of grain protein in wheat.	[58]
Heat stress	Maize	Plants were moved to a greenhouse for heat stress treatment (35.0 °C in the day) after artificial pollination.	Grain weight and starch deposition were suppressed by heat stress due to the decreased activities of enzymes involved in starch synthesis, and the increased protein content was due to the enhanced activity of glutamate synthase.	[59]
Water stress	Rice	Drought stress with different soil water levels at the jointing–booting stage was set as the whole-plot factor with three levels: mild drought stress (−10 kPa), moderate drought stress (−25 kPa), and severe drought stress (−40 kPa).	Drought stress significantly reduced dry matter accumulation in the stems and leaves. Mild and moderate drought increased dry matter translocation efficiency. The decrease in dry matter translocation after anthesis directly caused a change in the grain-filling strategy. The grain-filling rate decreased significantly with an increase in the drought stress.	[55]
Water deficit stress	Rice	From 9 days after anthesis to maturity, two levels of soil water potential (well-watered, WW = 0 MPa; water deficit stressed, WS = −0.05 MPa) were imposed on plants.	WS enhanced remobilization of carbon from reserve pools to grains, increased the harvest index, shortened the grain filling period, and increased the grain filling rate at normal and high nitrogen levels.	[60]
Water deficit stress	Wheat	From 9 days after anthesis to maturity, three levels of soil water potential (well-watered, WW = −0.02 MPa; moderate water deficit, MD = −0.04 MPa; severe water deficit, SD = −0.06 MPa) were imposed on plants.	MD enhanced senescence by accelerating the loss of leaf nitrogen and chlorophyll and increasing lipid peroxidation. Soil drying shortened the grain filling period, but the grain filling rate was substantially increased by MD and SD treatments.	[61,62]
Nitrogen treatment	Rice	Three nitrogen treatments, applied at the stage of panicle initiation or spikelet differentiation or both, were adopted with no N application during the mid-season as a control.	Nitrogen application at the spikelet differentiation stage significantly increased NSC per spikelet at the heading time, sink strength, grain filling rate, and grain weight of inferior spikelets in super rice, whereas N application at the panicle initiation stage or at both the panicle initiation and spikelet differentiation stages significantly reduced these.	[63]
Excessive nitrogen treatment	Rice	Normal nitrogen (NN) and high nitrogen (HN) application rates were 120 kg N ha^−1^ and 240 kg N ha^−1^ in the form of urea in a paddy field.	Excessive nitrogen application reduced the grain filling rate and grain weight of inferior spikelets but not those of superior spikelets by suppressing the accumulation of starch.	[19]
Salt stress	Rice	Application of 0.75% NaCl to a salt-sensitive rice cultivar at late booting	Spikelets per panicle and the percentage of filled grain decreased significantly in response to NaCl application. Application of 0.75% NaCl to a salt-sensitive rice cultivar at late booting resulted in a >20% yield loss, and the inhibitory effect on grain filling was greater in basal than in apical spikelets.	[15]
Saline–alkali stress	Rice	Saline–alkali stress (0.6% sea salt solution, pH 7.5 ± 0.3) was started from 6 to 30 days after anthesis by using sundried sea salt dissolved in tap water before treatment and was maintained at a water depth of 5–8 cm.	Saline–alkali stress significantly reduced the grain weight by decreasing the accumulation of starch and NSC in grains.	[16]

There are a variety of dose–response models, mainly including the linear non-threshold stress response model, threshold stress response model, and hormesis stress response model (Figure 1) [46,64,65]. The former two have long dominated the field of dose–response research, while the latter one was brought into mainstream research more recently [46]. Most of the environmental stress dose–response relationships relating to grain filling belong to the linear non-threshold stress responses and threshold stress responses (Figure 1). At high doses beyond the threshold, adverse effects appear, including low grain filling rates, poorly filled grains, and reduced crop yield and quality. There are, of course, exceptions. Water stress shows a hormesis stress response. Severe drought limits crop production and reduces the yield and quality of cereals [53,55], while moderate soil drying was found to induce hormesis. Mild soil drying without affecting photosynthesis can enhance whole-plant senescence, leading to an increased grain filling rate and grain weight [3,56,57]. Furthermore, alternate wetting and moderate soil drying significantly increased grain weight and grain yield while reducing chalky grains and chalkiness degree under high temperature [66,67].

## 3. Physiological, Molecular, and Biochemical Basis of Grain Filling Triggered by Environmental Factors

Environmental factors are contributory factors that determine grain filling (Table 1). Much is now known about the environmental factors that play important roles in grain filling and about how these environmental factors act in grain filling.

### 3.1. Light as an Important Factor That Maintains the Activity of Starch Synthesis Enzymes in Kernels

Light affects plant morphogenesis and regulates the carbon metabolism of crops by influencing the photosynthetic rate, providing the necessary energy and organic building blocks for almost all living beings on planet Earth. Low light during the developmental stage substantially decreases leaf Pn, Gs, and chlorophyll content and hinders carbon assimilation [17]. Another study also indicated that low light limits the diffusion of carbon dioxide from the atmosphere into the leaves, resulting in reductions in Gs and Pn [68]. Over 90% of crop biomass is derived from photosynthetic products [69], stored in the stem and leaves, and finally transported into the kernels to form starch [3].

Furthermore, light and the activities of enzymes involved in regulating starch synthesis were shown to be relevant. Low light stress reduces the activities of GBSS and SBE during grain filling, resulting in reduced starch accumulation, as well as a reduction in yield [70]. Most recently, it was proved that low light stress significantly decreases the activity of AGPase, SuS, and SS at the early grain filling stage. Moreover, the activities of GBSS and SBE are also suppressed in the late period of grain filling, leading to slower grain filling and poorer grain fullness [18]. Therefore, low light not only interferes with the photosynthate supply for grain filling but also decreases starch-synthesis-related activity, thus having compounding effects on grain filling.

### 3.2. Temperature: Moderation Is Best

The optimum daily temperature for grain filling ranges from 22 to 27 °C for rice [71] and from 17 to 23 °C for wheat [54]; temperatures outside of these ranges have negative effects on grain filling.

On the one hand, extreme temperatures decrease the overall photosynthetic rate and reduce carbohydrate accumulation [72]. Ribulose-1,5-bisphosphate carboxylase/oxygenase acts as a key enzyme in regulating photosynthesis response to heat and low temperature stress [73,74]. Extreme temperatures lead to the blocked synthesis of photosynthetic pigments and membrane disruption, thus inhibiting the activities of enzymes and destroying photosynthetic systems [75,76]. Furthermore, high and low temperature stress significantly reduce the leaf area in rice [77] and wheat [78]. Ultimately, extreme temperatures reduce both the photosynthesis rate and carbohydrate accumulation. On the other hand, extreme temperatures decrease the key enzymes for starch synthesis, including AGPase, SS, SuS, GBSS, and SBE [79,80], resulting in chalkiness of the rice grain and reduced starch accumulation, especially under high night time temperatures after anthesis [81]. In maize grains, heat stress could interfere with sucrose degradation to hexose and reduce starch biosynthesis in the endosperm by reducing vacuolar invertase activity [82].

According to a study by Yamakawa et al. [83], under high temperatures, the down-regulation of several genes for starch or storage protein synthesis and up-regulation of genes for starch-degrading α-amylases are the main reasons for the inhibition of starch accumulation and chalkiness of the rice grain. The soluble starch synthase I (*SSI*) gene is important for amylopectin synthesis, while overexpression of the rice *SSI* gene in transgenic wheat can significantly improve wheat productivity under heat stress, with the thousand-kernel weight increased 21–34% compared with that of control plants [84]. *OsbZIP58* could bind to promoters of and regulate the expression of storage compound synthesis genes, including *OsAGPL3*, *Wx*, *OsSSIIa*, *SBE1*, *OsBEIIb*, *α-globulin*, and *ISA2* [85,86]. OsbZIP58β with lower transcript activity by alternative splicing was induced by high temperature, resulting in floury and shrunken endosperms and dramatically reduced storage materials in the seeds [87]. Furthermore, impeded sucrose transportation might contribute to grain weight reduction and grain chalkiness. Seed-specific NAM/ATAF/CUC (NAC) domain transcription factors, *ONAC127* and *ONAC129*, are responsive to heat stress and regulate grain filling by binding promoters of and regulating the expression of sugar transporters *OsMST6* and *OsSWEET4* [88]. *OsSUT1* encodes a rice sucrose transport protein that is highly expressed in developing grain, and antisense suppression of *SUT1* decreased the grain weight in the transgenic rice plant [89]. The expression of *SUT1* was down-regulated by high-temperature ripening, suggesting that high temperature may influence grain filling by affecting sucrose transportation.

### 3.3. Water Conditions: Signal for Carbohydrate Remobilization

Water is essential for plant growth and is the most limiting factor for plant growth and reproduction. Periods of water demand in cereals vary in their timing and intensity, but available water is critical during grain filling [90]. However, water conditions are a double-edged sword for grain filling. On the one hand, water stress occurring during early grain development reduces the number of endosperm cells and amyloplasts formed, thus curtailing the kernel sink potential and reducing the capacity for starch accumulation, finally leading to a reduction in grain weight [91,92]. On the other hand, mild soil drying after anthesis can significantly promote grain filling in many crops, such as rice and wheat. This is achieved primarily by increasing the remobilization of nonstructural carbohydrates (NSC) from the vegetative tissues to the grain and accelerating the grain filling rate by enhancing the activities of key enzymes in the sucrose-to-starch catalytic pathway. The physiological mechanisms of grain filling of cereals under soil drying were reviewed in our previous review paper [3,93].

In recent years, the molecular basis of grain filling induced by moderate water stress has been investigated, and most importantly, several key genes and their regulatory networks were identified [56,94,95,96,97,98]. In the stem, the expression of proteins involved in starch degradation and the sucrose re-synthesis pathway was enhanced by moderate soil drying, including β-glucosidase and starch phosphorylase (SPS8 and SPP1). In addition, some transporters related to carbon reserve remobilization, like monosaccharide transporters and sucrose transporters, were also upregulated in the stem under moderate soil drying [99]. Sugars Will Eventually Be Exported Transporter6b (*SWEET6b*), an essential gene in carbon reserve remobilization in rice stems and grains, was demonstrated to be more highly methylated with elevated transcript levels under moderate soil drying [95]. Most recently, a field experiment demonstrated that post-anthesis mild soil drying can enhance source-to-sink remobilization of nitrogen and synergistically increase grain yield and nitrogen use efficiency via redistributing cytokinin (CK) from source to sink in rice [98].

Apart from the regulation of gene expression in the stem, the activities of the enzymes and the expression of the genes involved in hormone concentrations/ratios and the sucrose–starch metabolic pathway in the grains under mild soil drying also contribute to grain filling. Plant hormones, such as abscisic acid (ABA), gibberellic acids (GA), auxin (IAA), and CK, are involved in controlling grain filling in rice and wheat under soil drying [56,100]. Most recently, our results showed that downregulation of the gene *OsABA8ox2* under moderate soil drying resulted in increased ABA content in the inferior spikelets, leading to increased enzyme activities and being responsible for the conversion of sucrose into starch [56,94]. Meanwhile, the IAA biosynthesis genes *OsYUC11* and *OsTAR2* were upregulated by moderate soil drying and exogenous ABA application, leading to elevated IAA content in the inferior spikelets. All the results above suggest that the synergistic interaction of ABA-mediated accumulation of IAA promotes grain filling of inferior spikelets under moderate soil drying [56]. The dynamic expression of various miRNAs during grain filling suggests the involvement of miRNA in regulating this fundamental process [101]. Many miRNAs are differentially expressed under moderate soil drying treatment, some of which have been reported to function in regulating grain size and grain weight. Notably, three miRNA target pathways—the miR1861-*OsSBDCP1*-*OsSSIIIa* pathway, miR397-*OsLAC* module, and miR1432-*OsACOT* pathway—regulate grain filling in inferior spikelets of rice by both starch synthesis and phytohormone biosynthesis under mild soil drying [96]. Like miRNAs, alternative splicing (AS), another post-transcriptional regulation mechanism, was identified in inferior spikelets under moderate soil drying treatment. Some of the splicing factors and starch-synthesis-related genes, like SR protein, hnRNP protein, *OsAGPL2, OsAPS2, OsSSIVa, OsSSIVb, OsGBSSII*, and *OsISA1*, showed differential AS changes under moderate soil drying, which provides a potential novel approach for the regulation of grain filling in rice under moderate soil drying [97]. Based on the results of the above studies, the regulatory mechanisms and genes are good resources to be used in rice breeding to increase grain weight.

### 3.4. Fertilizer: Too Much of a Good Thing

Fertilizers are substances used to improve crop growth and yield, and they are also a guarantee to achieve a good harvest, particularly nitrogen fertilizers. Nitrogen application at the young panicle differentiation stage could increase not only the spikelet number but also the ratio of NSC to spikelets, thus enhancing the sink strength and grain filling of both superior and inferior spikelets in rice [63]. However, there can be too much of a good thing. While overuse of nitrogen will increase the grain yield to some extent, the grain filling rate and quality of rice grains, as well as the nitrogen use efficiency (NUE), are significantly reduced by excessive nitrogen [19,102]. Meanwhile, there are side effects on the environment, such as environmental pollution and eutrophication of water [103,104]. After years of research, the physiological and molecular mechanisms of excessive nitrogen in grain filling are gradually becoming better understood.

On the one hand, the delay in the process of plant senescence caused by overuse of nitrogen fertilizer will retard the remobilization of nutrients from vegetative tissues to grains, lead to a low NUE and grain filling rate, and eventually result in many poorly filled grains [3,105]. On the other hand, changes in key enzymes for starch biosynthesis and concentrations/ratios of plant hormones under the overuse of nitrogen fertilizer are also associated with grain filling [19,98,100]. SuS, AGPase, and StS were significantly suppressed in inferior spikelets under high nitrogen application [19]. Under high nitrogen conditions, GA was enhanced in the grains during grain filling, which resulted in a reduction in the ABA-to-GA ratio [100]. This led to a low grain filling rate and poorly filled grains in rice [106]. Unlike GA, both CK and IAA were reduced by excessive nitrogen fertilizer mainly via the down-regulated expression of IAA biosynthesis genes and up-regulated expression of CK oxidase genes [19]. The application of exogenous IAA or CK to the panicle under excessive nitrogen fertilizer can significantly increase the enzyme activities of SuS, AGPase, and StS and enhance source-to-sink remobilization of nitrogen, increasing the grain yield and NUE [19,98]. Optimal application of nitrogen fertilizer is not only crucial in increasing grain yield and NUE but also a benefit to the environment, which should be widely promoted in agricultural production.

### 3.5. Saline–Alkali Stress: Ongoing Challenges in Worldwide Agriculture

Soil salinity is one of the major environmental stresses causing a decrease in grain weight, grain filling rate, grain number, and seed setting rate [15,16,107,108,109]. Firstly, salt stress causes a grain yield reduction mainly by limiting carbohydrate production driven by photosynthesis and the transport of this biomolecule to spikelets [110,111,112,113]. Secondly, genes of the starch biosynthesis pathway, including *OsSUS*, *OsAGPL*, *OsAGPS*, *OsSSI*, and *OsSSIIIa*, were found to be downregulated in saline–alkali-treated plants. In agreement, the activities of SuS, AGPase, StS were suppressed, which may be a possible reason for the poor grain filling in response to salt application [15,16]. One reason may be that *SALT-RESPONSIVE ERF1* (*SERF1*), involved in salt stress tolerance, negatively regulates grain filling by directly binding to the promotor of *GBSSI* (*Wx*) and by storage protein synthesis with *RICE PROLAMIN BOX BINDING FACTOR (RPBF)* in rice [114,115].

Plant hormones are critical in regulating plant growth and stress responses; the function of some of them in grain filling is discussed above. Unlike ABA, IAA, and CK, which promote grain filling, ethylene inhibits grain filling and displays a negative correlation with the grain filling process. Furthermore, exogenous spraying of ethylene significantly reduces grain filling [116,117]. In recent years, studies have shown that salt-induced inhibition of grain filling might be mediated by ethylene [15,16]. Several key genes involved in the ethylene biosynthesis and signaling pathway, including the *OsACS* and *OsACO* genes, were found to be upregulated under saline–alkali treatment [16]. Together, the increased ethylene production and activated ethylene signaling and the salt-inhibited activities of starch-metabolism-related enzymes in rice grains are the main reason for salt-induced poor grain filling.

## 4. Adaptation Strategies in Response to Environmental Stimuli

In the face of the increasing food demand and the trend of environmental deterioration, it is extremely urgent to adjust crop structures and determine appropriate agricultural practices. Different approaches like modern breeding and better crop management practices are used to help cereals cope with the changing climate [1,118,119,120,121,122]. Thus, in order to adapt cereals to environmental stress conditions and have better grain filling under environmental stimuli, the following approaches are mandatory.

### 4.1. Selection of Resistant Varieties and Plant Breeding

The breeding of improved and resistant varieties is a key strategy to adapt crops to climate change, and it is also a viable way to minimize losses in cereal production. Crop varieties originated through selection during early human civilization, which led to better quality, higher yield, and enhanced environmental adaptability [123]. When faced with growing environmental deterioration, these same principles apply. Indeed, much has been done, including the development of some resistant varieties [124,125,126,127,128,129,130,131]. Improved varieties with resistance and tolerance to environmental stresses for cereal crops can be applied broadly in agricultural production in various countries.

By comparing the responses of different varieties under adversity, we can not only obtain resistant varieties but also identify QTLs or genes which can be utilized for marker-assisted resistance breeding. For example, Zhang et al. [132] obtained a heat-tolerant variety (African rice CG14) and a heat-sensitive variety (Wuyunjing) through cultivar screening and identified a quantitative trait locus, *Thermo-tolerance 3* (*TT3*), consisting of two genes, *TT3.1* and *TT3.2*, which interact with each other to enhance rice thermotolerance and reduce grain yield losses caused by heat stress. Overexpressing *TT3.1* or knocking out *TT3.2* conferred significant yield increases under heat stress, providing a strategy for breeding heat-tolerant crops. Saito et al. mapped a quantitative trait locus for cold tolerance, *Ctb1*, from a cold-tolerant variety (Norin-PL8) [133]. When *Ctb1* was introduced into a cold-sensitive variety, the degree of spikelet fertility and cold tolerance were significantly improved [134]. Overexpression of RBG1 enhanced tolerance to heat, osmotic, and salt stresses, as well as rapid recovery from water deficit stress [130]. Through transcriptome analyses, Wei et al. found that *ZmbHLH124* is expressed differently between drought-tolerant (RIL70 and RIL73) and -sensitive lines (RIL44 and RIL93). Overexpression of *ZmbHLH124* in the drought-tolerant line in maize and rice enhanced plant drought tolerance [131]. These studies represent a valuable resource for resistance breeding.

### 4.2. Crop Management Practices

Improved crop management practices can minimize the negative impacts of an adverse environment and decrease crop production loss. Firstly, changing the sowing date to evade the harmful influences of an adverse environment during the flowering and early grain filling stages has been suggested as an adaptation tactic. For example, postponing the sowing date of single-season rice in the middle and lower reaches of the Yangtze River could avoid the high temperatures during late August [125]. However, for double-season rice planting areas, earlier sowing of late rice could avoid low temperatures, overcast skies, and rain. Similarly, it is also useful to escape hot and dry periods in the growing seasons by earlier sowing of winter and spring cereal crops and delayed sowing of summer and autumn cereals [1]. Inclusion of the regional environment in decision making and greater consideration of genotype–environment interactions can be used to identify sowing dates that are best adapted to the specific local environment.

Secondly, we can reduce the exacerbating effects of an adverse environment through good agronomic management. For example, supplementary irrigation during stress-sensitive periods, which can avoid crop exposure to lethal heat and drought stresses, and frost in cooler areas, can enhance grain yield [135,136]. Deep-flood irrigation or continuous irrigation with running water is effective in decreasing the soil and water temperature and increasing the available assimilate supply per grain, thus leading to improved grain quality [137,138]. However, many farmers cannot afford abundant or controlled irrigation. In this context, one promising example is presented by Duan et al.: alternate wetting and moderate soil drying significantly increased the seed setting rate, grain weight, and grain yield, and consequently reduced chalky grains and chalkiness degree under high temperature [66,67]. Similarly, parental drought priming plants had a higher ATP concentration and higher activity levels of the enzymes involved in sucrose biosynthesis and starch metabolism, leading to induced low-temperature tolerance in offspring [139]. Soil mulching, such as straw mulching and ridge-furrow plastic film mulching can improve maize yield in semi-humid drought-prone regions [140]. In addition to water management, a rational fertilization level also plays a key role in improving crop tolerance [19]. Meanwhile, a rationally increasing amount of fertilizer application may also promote plant growth and mitigate the adverse effects of climate change [141]. Increased nitrogen application at the late growth stage may alleviate the negative effects on grain filling caused by high temperature [142,143,144,145]. Compared to inorganic amendment, poultry manure amendment had a greater ameliorative capacity for grain yield under heat and moderate water stress [146]. An optimized planting pattern with increased number of seedlings per hill and a reduced number of hills can optimize the canopy structure and light utilization, leading to decreased chalky grain rate and chalkiness degree under shading stress [147]. The improved wide–narrow row planting pattern is also a popular way to optimize the canopy light environment [148]. It follows that good agronomic management is very important.

Thirdly, exogenous application of chemical compounds can be employed to mitigate adverse environmental effects on grain filling. For example, exogenous silicon alleviates the spikelet fertility reduction in hybrid rice induced by high temperature [149]. The same effect was reported when spraying micronutrient fertilizers containing zinc nutrition, KH_2_PO_4_, α-tocopherol and glycine betaine, ascorbic acid, spermidine, brassinosteroids, and methyl jasmonates under high temperature conditions [67,150,151,152,153]. Similarly, the application of exogenous brassinosteroids mediated the effect of soil drying on spikelet degeneration [154] and improved the photosynthetic performance of maize under combined mercury and drought stress [155]. Exogenous IAA or CK treatment can also make up for the reduction in inferior grain weight induced by excessive nitrogen application [19]. Meanwhile, the application of exogenous ABA and CK can overcome the adverse effects of unfavorably delayed whole-plant senescence caused by the heavy use of nitrogen fertilizers, enhancing the remobilization and transfer of assimilates, and promoting grain filling [3,156,157].

Thus, it is time to establish smart agriculture combining regional environment considerations, cultivar selection, disaster warnings, and cultivation technology, which is the trend of future agricultural practices.

## 5. Conclusions and Future Perspectives

Grain filling, as an important growth process, is very sensitive to environmental variations, especially in the inferior spikelets. As shown in Figure 2, environmental stimuli affect grain filling in three main ways. First is a direct decrease in photosynthetic parameters and consequent limitations of carbohydrate accumulation in both the leaf and stem. Second is the regulation of the key enzymes involved in carbon remobilization, thereby limiting the transport of NSC from leaves and stems to spikelets. Thirdly, the key enzymes involved in sugar–starch metabolism in grains are also inhibited under environmental stimuli directly and indirectly through phytohormones. Therefore, future research should focus on investigating the aspects that we can evaluate the impact of environmental stimuli on grain filling and make recommendations to mitigate environmental impacts.

While efforts to implement sustainable agricultural development have led to an improved agricultural environment, treatment outcomes remain unsatisfactory. Adaptation strategies are the most important way to reduce the negative effects of environmental stimuli on grain filling in cereals. We recommend that different adaptation approaches, including the selection of resistant varieties, modern breeding and genetic modifications, better and improved crop management, and the application of exogenous chemical compounds, should be used to increase plant tolerance for better growth and grain filling against to unfavorable environmental conditions. Meanwhile, future endeavors should also pay attention on addressing the growing demand for smart agriculture combining regional environment considerations, cultivar selection, disaster warnings, and cultivation technology. Moreover, we advance the gene function-related research and the commercialization of gene-edited crops to improve sustainable agricultural practices. All in all, there is still much work to do in order to tackle future food security challenges under increasing environmental degradation.

## Figures and Tables

**Figure 1 ijms-24-02255-f001:**
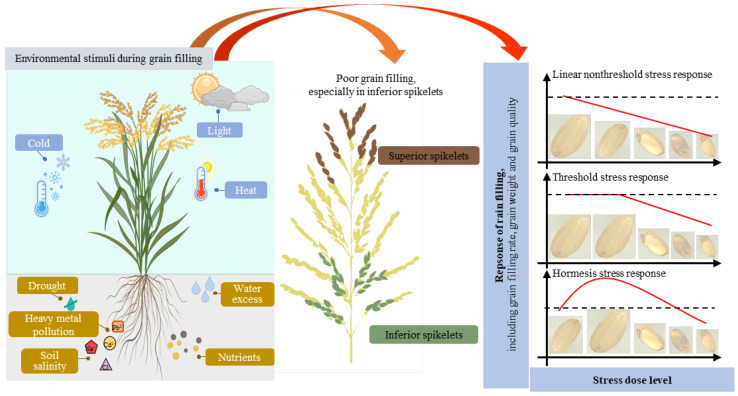
Environmental stimuli and cereal response during grain filling. Cereals are continuously exposed to different environmental stimuli (e.g., light, heat, cold, drought, and soil salinity) above- and belowground. Stress dose–response relationships in grain filling are categorized based on their responses to the stress dose level into linear nonthreshold stress response, threshold stress response, and hormesis stress response [45,46]. The dashed line indicates the control. This figure was created with the help of Figdraw (www.home-for-researchers.com, accessed on 1 November 2022).

**Figure 2 ijms-24-02255-f002:**
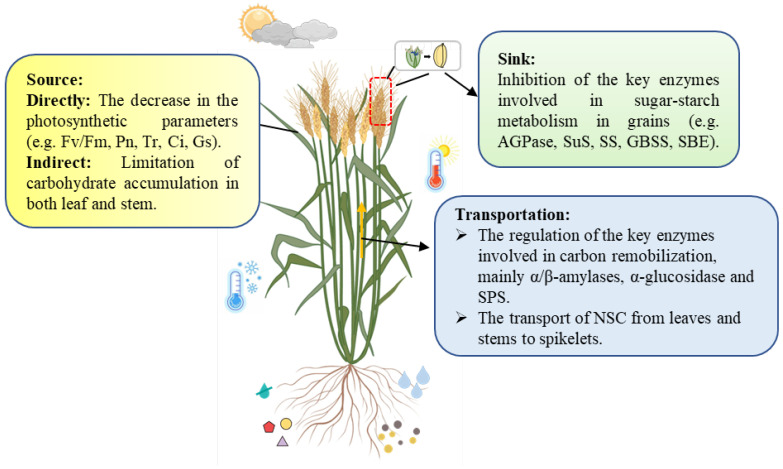
The effect of environmental stimuli on sources, transportation, and sinks during grain filling. Fv/Fm: photosynthetic efficiency; Pn: photosynthetic rate; Tr: transpiration rate; Ci: intracellular CO2 concentration; Gs: stomatal conductance; AGPase: ADP-glucose pyrophosphorylase; SuS: sucrose synthase; SS: soluble starch synthase; GBSS: granule-bound starch synthase; SBE: starch branching enzyme; SPS: sucrose-phosphate synthase. This figure was created with the help of Figdraw (www.home-for-researchers.com, accessed on 1 November 2022).

## Data Availability

All of the data generated or analyzed during this study are included in this published article.

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
