# Peer review of "Environmental Stimuli: A Major Challenge during Grain Filling in Cereals"

_ijms, 2023, doi:10.3390/ijms24032255_

Round 1

Reviewer 1 Report

The review paper should be concisely written withou repetition. You have too much repetition. Manuscript is hard for reading. Two Figures are not enought for serious review paper. Figures are only graphical presentation of already written text. There is no Tables that would summarize all previous results about topic. Tables would improve text significantly (authors have too much text, hard to read. Instead, make Tables for all data and discuss about in the text). All manucript is compilation of some previous work without disscusion and personal opinion. All facts, stimuli and environmental impact were already well known. In conclusion: my major remark is absence of Tables, too much repetition, and missing of point.... Everything you write has already been applied in agriculture and you have to give you personal opinion for further perspectives.

Grain filling-explain more what is grain filling at the beginning of introduction

44 erase ecosystems, End of the sentence

44 Factors affecting grain filling are low light......

46 are urgent

50 Therefore, both carbon sources are affected and hindered by environmental stimuli

51 Drought and heat can reduce tha accumulation of certain seed components, mainly starch and proteins by suppression of enzymatic reactions (starch and protein synthesis) and in that way these factors have stron influence on grain weight and quality.

52 which enzymatic processes? Starch synthesis?

54 soil stress inhibits activity of starh synthesis enzymes? That is the same as drought and heat, not unlike.

60 too much nitrogen results in delayed senescence...

63....spikelets, and therefore will improve the grain weight.

64 Mild soil drying can also enchance plant senescence and faster and better remobilzation of carbon from reserve pools to grains leading to faster grain filling of both superior and inferior spikelets.

67 it can be concluded that these factors...

69 The physiological and molecular mechanism of grain filling have been studied intensively in recent years, especially under different environmental stimuli. In this work, we described the current status of agricultural environment and stress dose responce of grain filling in cerels. We also illustrated how.....grain filling. At the end of review we gave some advices....We hope that these efforts would provide.....

I have tried to improve English so far, but I will not deal with it any further. I will highlight only important remarks.

79 Environmental stimuli influence on grain filling in cereals

I recommend the following subtitles: Light, Temperature, Water, Soil...for every stimulus you described

81 An adequate environment is required for normal plant growth

88-91 rephrase the sentence

96 subtitle Temperature

124 Figure 1. Cereals are continuously exposed to different environmental stimuli(...).Stress dose...The interrupted line indicates control.

3.1. 178-200 light was already described partially in the section 85-95. . Every environmental stimuli should be at the same place. If you split text on many places, that is too difficult to follow. Same for temperature and other stimuli.

In this subtitle lines 79-130 there is too much repetition. Sentence like line 94, 96, 109, 116, 122 all explain the same fact that is bad impact on agriculture. This fact could be written at the beginning, not in every paragraph.

132 repetition

138 you already said about light, temperature, water and fertilizer. Too much repetition

Subtitle 2 and subtitle 3 are the same. Maybe subtitle 3: Molecular (physiological, biochemical) mechanism basis of grain filling triggered by environmental factors

174-177 repetition

3.1 Light as important factor that maintains activity of starch synthesis enzymes in kernels

180-189 repetition

202-205 repetition

239 3.3. Water: Signal for carbohydrates remobilization

240-250 repetition. Some parts should be in previous section because there is no connection with molecular mechanisms

266-270 plant hormones are not part of this section

294-304 repetition

322 3.5. Saline-alkali stress: ongoing challenges in worldwide agriculture

335-345 more plant hormones… not part of this section. Author may describe the role of plant hormone in grain filling as a separate subtitle.

354-374 add some selected varieties, their cultivar names about new cultivar selected for some other stresses, not only for temperature. You described only cultivar resistant to cold and heat. There are more cereals varieties resistant to dry soil, lack of water (all stresses you described in this work)

Figure 2 is graphical presentation of text, already explained. Text below Figure was already written. Repetition

444-459 repetition

Author Response

Response to Reviewer 1:

The review paper should be concisely written withou repetition. You have too much repetition. Manuscript is hard for reading. Two Figures are not enought for serious review paper. Figures are only graphical presentation of already written text. There is no Tables that would summarize all previous results about topic. Tables would improve text significantly (authors have too much text, hard to read. Instead, make Tables for all data and discuss about in the text). All manucript is compilation of some previous work without disscusion and personal opinion. All facts, stimuli and environmental impact were already well known. In conclusion: my major remark is absence of Tables, too much repetition, and missing of point.... Everything you write has already been applied in agriculture and you have to give you personal opinion for further perspectives.

Response to comments:

Many thanks for the comment. We have carefully revised our manuscript according to the constructive comments from the reviewers and provided point-by-point responses. We agree that our presentation of this issue involved some unnecessary repetition and have amended the manuscript to improve this. A table has been added to provide more focus in the new manuscript. We believe that our manuscript is substantially improved and hope that the revised manuscript is now suitable for publication.

Grain filling-explain more what is grain filling at the beginning of introduction

Response to comments:

Grain filling is the processes of cell proliferation and cell expansion, as well as the processes of photoassimilate interconversion and starch accumulation. We have added this explanation in the revised manuscript.

44 erase ecosystems, End of the sentence

44 Factors affecting grain filling are low light......

46 are urgent

50 Therefore, both carbon sources are affected and hindered by environmental stimuli

51 Drought and heat can reduce tha accumulation of certain seed components, mainly starch and proteins by suppression of enzymatic reactions (starch and protein synthesis) and in that way these factors have stron influence on grain weight and quality.

Response to comments:

Thanks for the suggestion. we have corrected them in the new manuscript.

52 which enzymatic processes? Starch synthesis?

Response to comments:

Thanks a lot for your comment. It should be starch and protein synthesis

54 soil stress inhibits activity of starh synthesis enzymes? That is the same as drought and heat, not unlike.

Response to comments:

We are very sorry for the mistake in the confused description. It should be OsFCA gene. We have corrected them in the new manuscript.

60 too much nitrogen results in delayed senescence...

63....spikelets, and therefore will improve the grain weight.

64 Mild soil drying can also enchance plant senescence and faster and better remobilzation of carbon from reserve pools to grains leading to faster grain filling of both superior and inferior spikelets.

67 it can be concluded that these factors...

69 The physiological and molecular mechanism of grain filling have been studied intensively in recent years, especially under different environmental stimuli. In this work, we described the current status of agricultural environment and stress dose responce of grain filling in cerels. We also illustrated how.....grain filling. At the end of review we gave some advices....We hope that these efforts would provide.....

I have tried to improve English so far, but I will not deal with it any further. I will highlight only important remarks.

Response to comments:

Many thanks for the comment, we have corrected them as suggested in the new manuscript. The revised manuscript has now been carefully proofread and polished by an English-editing professional service.

79 Environmental stimuli influence on grain filling in cereals

I recommend the following subtitles: Light, Temperature, Water, Soil...for every stimulus you described

Response to comments:

Thank you for your suggestions, and we have corrected them in the new manuscript.

81 An adequate environment is required for normal plant growth

88-91 rephrase the sentence

96 subtitle Temperature

124 Figure 1. Cereals are continuously exposed to different environmental stimuli(...).Stress dose...The interrupted line indicates control.

Response to comments:

Thank you for your suggestions, and we have corrected them in the new manuscript.

3.1. 178-200 light was already described partially in the section 85-95. . Every environmental stimuli should be at the same place. If you split text on many places, that is too difficult to follow. Same for temperature and other stimuli.

Response to comments:

Thanks for the comment. The propose of the section 2.1 is to show that the current status of agricultural production is taking on environmental stimuli (light, temperature, water, soil). Section 3 focus on the mechanism of action of environmental factors on grain filling.

In this subtitle lines 79-130 there is too much repetition. Sentence like line 94, 96, 109, 116, 122 all explain the same fact that is bad impact on agriculture. This fact could be written at the beginning, not in every paragraph.

132 repetition

138 you already said about light, temperature, water and fertilizer. Too much repetition

Response to comments:

Thanks for your suggestions, and we have altered or deleted the repetitive statements in the new manuscript.

Subtitle 2 and subtitle 3 are the same. Maybe subtitle 3: Molecular (physiological, biochemical) mechanism basis of grain filling triggered by environmental factors

Response to comments:

Many thanks for the comment. Yes, this is a good suggestion, and we have changed it to” Physiological, molecular, and biochemical basis of grain filling triggered by environmental factors”.

174-177 repetition

Response to comments:

Thanks for your suggestions, and we have altered or deleted the repetitive statements in the new manuscript.

3.1 Light as important factor that maintains activity of starch synthesis enzymes in kernels

Response to comments:

Thanks for your suggestions, and we have made the suggested changes.

180-189 repetition

202-205 repetition

Response to comments:

Thanks for your suggestions, and we have altered or deleted the repetitive statements in the new manuscript.

239 3.3. Water: Signal for carbohydrates remobilization

Response to comments:

Thanks for your suggestions, and we have made the suggested changes.

240-250 repetition. Some parts should be in previous section because there is no connection with molecular mechanisms

266-270 plant hormones are not part of this section

Response to comments:

Many thanks for the comment. We have changed subtitle 3 to” Physiological, molecular, and biochemical basis of grain filling triggered by environmental factors”. I think this part belongs to this section.

294-304 repetition

322 3.5. Saline-alkali stress: ongoing challenges in worldwide agriculture

Response to comments:

Thanks for your suggestions, and we have made the suggested changes.

335-345 more plant hormones… not part of this section. Author may describe the role of plant hormone in grain filling as a separate subtitle.

Response to comments:

Many thanks for the comment. We have changed subtitle 3 to” Physiological, molecular, and biochemical basis of grain filling triggered by environmental factors”. I think plant hormones belongs to biochemical mechanisms.

354-374 add some selected varieties, their cultivar names about new cultivar selected for some other stresses, not only for temperature. You described only cultivar resistant to cold and heat. There are more cereals varieties resistant to dry soil, lack of water (all stresses you described in this work)

Response to comments:

Many thanks for the comment. Following your suggestion, we have selected more cereals resistant varieties.

Figure 2 is graphical presentation of text, already explained. Text below Figure was already written. Repetition

444-459 repetition

Response to comments:

Thanks for your suggestions, and we have altered or deleted the repetitive statements in the new manuscript.

Reviewer 2 Report

Comments:

Environmental Stimuli: A Major Challenge during Grain Filling in Cereals

This is a very interesting work which mainly highlighted major environmental stimuli like Light, temperature, water and fertilizers regulating crop growth and productivity. In my opinion this paper briefly provide very interesting up to date concepts of the environmental stresses influencing grain filling in cereals and could direct new developmental ideas for future research. However, before going further to publication, I have some suggestion;

Section 2.2. Stress dose–responses of grain filling. It’s better to provide the summarized literature in a table having the species names, stress type, dose/concentration, Results/Impact on crop. See the given paper for reference;

https://doi.org/10.1016/j.tplants.2021.11.015

I have no other suggestions.

Author Response

Reviewer 2

This is a very interesting work which mainly highlighted major environmental stimuli like Light, temperature, water and fertilizers regulating crop growth and productivity. In my opinion this paper briefly provide very interesting up to date concepts of the environmental stresses influencing grain filling in cereals and could direct new developmental ideas for future research. However, before going further to publication, I have some suggestion;

Section 2.2. Stress dose–responses of grain filling. It’s better to provide the summarized literature in a table having the species names, stress type, dose/concentration, Results/Impact on crop. See the given paper for reference;

https://doi.org/10.1016/j.tplants.2021.11.015

I have no other suggestions.

Response to comments:

Many thanks for the comment. This is a very good advice. A table has been added.

Reviewer 3 Report

The manuscript by Teng Z. et al. “Environmental stimuli: A major challenge during grain filling in cereals” provides review on the data of the most important environmental factors action on crop growth and productivity. The manuscript excellently merits publishing in IJMS. It is written in good English, but contains some inaccuracies throughout the text.

1) Keywords section contains “1. Introduction” misplaced.

2) References to literary sources missed spacebars between sentence and square brackets throughout the text. Multiple references contrary possesses excess spacebars.

3) Line 105: Celsius degree sign contains an extra underscore.

4) Line 153: “infer” of “inferior” spikelets?

5) Line 182: “planet earth” – Earth has to be started with capital.

6) Lines 265 and later: commonly used abbreviation for cytokinin is CK, not CTK.

7) Lines 364-370: Does reference 125 correspond to Zhang et al? If so, it is better to put citation early, closer to “Zhang et al”.

8) Line 418: α-tocopherol and alpha-tocopherol twice mentioned in one sentence and in different forms (α- and alpha-). Does it make sense? And if so, it is better to unify.

Taking all mentioned above into account, the manuscript could be accepted after minor revision.

Author Response

Reviewer 3

The manuscript by Teng Z. et al. “Environmental stimuli: A major challenge during grain filling in cereals” provides review on the data of the most important environmental factors action on crop growth and productivity. The manuscript excellently merits publishing in IJMS. It is written in good English, but contains some inaccuracies throughout the text.

  • Keywords section contains “1. Introduction” misplaced.

2) References to literary sources missed spacebars between sentence and square brackets throughout the text. Multiple references contrary possesses excess spacebars.

3) Line 105: Celsius degree sign contains an extra underscore.

4) Line 153: “infer” of “inferior” spikelets?

Response to comments:

Sorry for the carelessness. we have corrected them in the new manuscript.

5) Line 182: “planet earth” – Earth has to be started with capital.

6) Lines 265 and later: commonly used abbreviation for cytokinin is CK, not CTK.

Response to comments:

Thanks for the suggestion. we have corrected them in the new manuscript.

7) Lines 364-370: Does reference 125 correspond to Zhang et al? If so, it is better to put citation early, closer to “Zhang et al”.

Response to comments:

Yes. we have corrected them in the new manuscript.

8) Line 418: α-tocopherol and alpha-tocopherol twice mentioned in one sentence and in different forms (α- and alpha-). Does it make sense? And if so, it is better to unify.

Response to comments:

Sorry for the mistake. α-tocopherol and alpha-tocopherol were the same substance. We have deleted “alpha-tocopherol” in the new manuscript.

Taking all mentioned above into account, the manuscript could be accepted after minor revision.

Response to comments:

Many thanks for the comment. We have carefully revised our manuscript according to the constructive comments from the reviewers and provided point-by-point responses. We believe that our manuscript is substantially improved and hope that the revised manuscript is now suitable for publication.